# Molecule Joint Auto-Encoding:
# Trajectory Pretraining with 2D and 3D Diffusion

**Weitao Du[1,3]** *    **Jiujiu Chen[2,3]**    **Xuecang Zhang[3]**    **Zhiming Ma[1]**    **Shengchao Liu[4]** †

[1] Department of Mathematics, Chinese Academy of Sciences
[2] Shenzhen Institute for Advanced Study, University of Electronic Science and Technology of China
[3] Huawei Technologies Ltd
[4] Department of Computer Science and Operations Research, Université de Montréal

## Abstract

Recently, artificial intelligence for drug discovery has raised increasing interest in both machine learning and chemistry domains. The fundamental building block for drug discovery is molecule geometry and thus, the molecule's geometrical representation is the main bottleneck to better utilize machine learning techniques for drug discovery. In this work, we propose a pretraining method for molecule joint auto-encoding (MoleculeJAE). MoleculeJAE can learn both the 2D bond (topology) and 3D conformation (geometry) information, and a diffusion process model is applied to mimic the augmented trajectories of such two modalities, based on which, MoleculeJAE will learn the inherent chemical structure in a self-supervised manner. Thus, the pretrained geometrical representation in MoleculeJAE is expected to benefit downstream geometry-related tasks. Empirically, MoleculeJAE proves its effectiveness by reaching state-of-the-art performance on 15 out of 20 tasks by comparing it with 12 competitive baselines. The code is available on this website.

## 1 Introduction

The remarkable progress in self-supervised learning has revolutionized the fields of molecule property prediction and molecule generation through the learning of expressive representations from large-scale unlabeled datasets [1–9]. Unsupervised representation learning can generally be categorized into two types [7, 10–13]: generative-based methods, encouraging the model to encode information for recovering the data distribution; and contrastive-based methods, encouraging the model to learn invariant features from multiple views of the same data. However, despite the success of large language models like GPT-3 [14, 15] trained using an autoregressive generation approach , learning robust representations from molecular data remains a significant challenge due to their complex graph structures. Compared to natural language and computer vision data [16, 17], molecular representations exhibit more complex graph structures and symmetries [18]. As molecular dynamics follows the principle of non-equilibrium statistical mechanics [19], diffusion models, as a generative method inspired by non-equilibrium statistical mechanics [20, 21], are a natural fit for 3D conformation generation. Previous researches have demonstrated the effectiveness of diffusion models, specifically 2D molecular graphs [22, 23] or 3D molecular conformers [24–27], for molecular structure generation tasks. However, a crucial question remains: *Can diffusion models be effectively utilized for jointly learning 2D and 3D latent molecular representations?*

---

*duweitao@mass.ac.cn

†shengchao.liu@umontreal.ca

37th Conference on Neural Information Processing Systems (NeurIPS 2023).

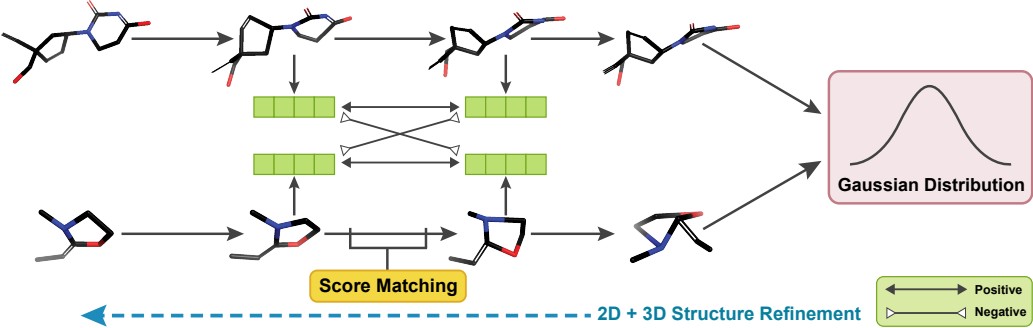

Figure 1: Pipeline of MoleculeJAE. For each individual molecule, MoleculeJAE utilizes the reconstructive task to perform denoising. For pairwise molecules, it conducts a contrastive learning paradigm to fit the trajectory.

Answering this question is challenging, given that diffusion models are the first generative models based on trajectory fitting. Specifically, diffusion models generate a family of forward random trajectories by either solving stochastic differential equations [28, 29] or discrete Markov chains [23, 30, 31], and the generative model is obtained by learning to reverse the random trajectories [32, 33]. It remains unclear how the informative representation manifests itself in diffusion models, as compared to other generative models that explicitly contain a semantically meaningful latent representation, such as GAN [34] and VAE [35]. In addition to the generative power of diffusion models, we aim to demonstrate the deep relationship between the forward process of diffusion models and data augmentation [36, 37], a crucial factor in contrastive learning. As a result, we ask whether a trajectory-based self-supervised learning paradigm that leverages both the generative and contrastive benefits of diffusion models can be used to obtain a powerful molecular representation.

**Our approach.** This work presents MoleculeJAE (Molecule Joint Auto-Encoding), a novel trajectory learning framework for molecular representation that captures both 2D (chemical bond structure) and 3D (conformer structure) information of molecules. Our proposed method is designed to respect the $SE(3)$ symmetry of molecule data and is trained by fitting the joint distribution of the data's augmented trajectories extracted from the forward process of the diffusion model. By training the representation in this manner, our framework not only captures the information of real data distribution but also accounts for the corelation between the real data and its noised counterparts. Under certain approximations, this trajectory distribution modeling decomposes into a marginal distribution estimation and a trajectory contrastive regularization task. This multi-task approach yields an effective and flexible framework that can simultaneously handle various types of molecular data, ranging from SE(3)-equivariant 3D conformers to discrete 2D bond connections. Furthermore, in contrast to diffusion models used for generation tasks that only accept noise as input, most downstream tasks have access to ground-truth molecular structures. To leverage this advantage better, we incorporate an equivariant graph neural network (GNN) block into the architecture, inspired by [38, 39], to efficiently encode crucial information from the ground-truth data. Analogous to conditional diffusion models [38, 40], the encoder's output serves as a meaningful guidance to help the diffused data accurately fitting the trajectory. In summary, our self-supervised learning framework unifies both contrastive and generative learning approaches from a trajectory perspective, providing a versatile and powerful molecular representation that can be applied to various downstream applications.

Regarding experiments, we evaluate MoleculeJAE on 20 well-established tasks drawn from the geometric pretraining literature [6, 7], including the energy prediction at stable conformation and force prediction along the molecule dynamics. Our empirical results support that MoleculeJAE can outperform 12 competitive baselines on 15 tasks. We further conduct ablation studies to verify the effectiveness of key modules in MoleculeJAE.

## 2   Background

In this section, we introduce the diffusion mechanism and relevant notations as a powerful data augmentation framework and elaborate on its instantiation in the context of molecular graph data.

## 2.1 Diffusion Mechanism as a trajectory augmentation

The concept of diffusion is widely used in various fields such as physics, mathematics, and computer science. In this paper, we take a broad view of diffusion by defining it as a Markov process with a discrete or continuous time index $t$, starting from a given data point $x_0$ and producing a series of random transformations that lead to $x_t$. If $t \in \{0, 1, 2, \dots\}$, then $x_t$ is a discrete Markov chain where the probability distribution $p(x_t)$ only depends on the distribution of $x_{t-1}$. We further refer to the conditional probability $p(x_t|x_{t-1})$ as the transition probability of the Markov chain. This definition enables us to model the evolution of data over time, allowing us to generate a sequence of augmented data points that span different temporal regimes.

In fact, we can adapt classical data augmentations, such as spatial rotation, color distortion, and Gaussian blurring [41, 42], as sources for defining $p(x_t|x_{t-1})$. For instance, Gaussian blurring generates a specific transition probability by setting $p(x_t|x_{t-1}) = \mathcal{N}(x_{t-1}, \epsilon)$, where the scaling factor $\epsilon$ may depend on time $t$.This transition probability rule perturbatively transforms the data for each step, and connecting all steps of transformed data produces a trajectory. In a similar vein, random masking can be seen as a discrete Markov chain, while continuous rotation can be viewed as a deterministic Markov process. By employing this approach, we can effectively expand the scope of one-step data augmentation operations to encompass trajectory-based scenarios.

In this paper, we build upon the concepts of generalized diffusion processes proposed in [33] to establish a unified framework for the trajectory augmentation examples mentioned earlier. Following the general approach in [33], each (cold) diffusion model is associated with a degradation Markov process $D(\cdot, t)$ indexed by a time variable $t \in [0, T]$, which introduces perturbations to the initial data $x_0$:

$$x_0 \xrightarrow{D(\cdot, t)} x_t. \tag{1}$$

Given that $x_0 \sim p_{data}$, the transformed variable $x_t := D(x_0, t)$, is also a random variable. To represent the marginal distribution of $D(\cdot, t)$ at time $t$, we use the notation $p_t(\cdot)$. When the context is clear, we will use the shorthand notation $D_t$ to refer to $D(\cdot, t)$. According to Equation (1), we define the pair $(x_0, x_t)$ as a multi-view of the sample $x_0$ at a specific time $t$. It is important to note that, under certain conditions, a Markov process converges to a stationary distribution that effectively erases any memory of the initial data $x_0$. Nevertheless, for small values of $t$, there is a high probability that $x_t$ remains within a ball centered around $x_0$.

In the context of the standard diffusion-based generative model [28, 32], the presence of a reverse process $R_t$ is necessary to undo the effects of the degradation $D_t$. In [28], the degradation $D(\cdot, t)$ is identified as the solution of a stochastic differential equation (SDE), and it has been demonstrated that a family of SDEs, including the probability flow ordinary differential equation (ODE) proposed in [28, 43], can reverse the degradation process $D(\cdot, t)$. Specifically, the inversion property implies that $R_{T-t}$ shares the same marginal distribution as $p_t$: $R_{T-t} \sim p_t$. Therefore, as $t$ varies from 0 to $T$, the reverse process $R_t$ gradually restores the data distribution. On the other hand, for cold diffusions, although the model is also trained by reconstructing the data from $D_t$, it does not explicitly assume the existence of a reverse process as rigorously as in the continuous diffusion-based generative model.

**Example of heat diffusion** We use the term 'heat' to describe a diffusion process that involves injecting Gaussian noise. Under this definition, the continuous limit of heat diffusion can be expressed by the solution of stochastic differential equations (SDEs):

$$dx_t = \mu(x_t, t)dt + \sigma(t)dw_t, \quad t \in [0, T] \tag{2}$$

Here, $w_t$ represents the standard Brownian motion. It is worth noting that while the solution $x_t$ of a general SDE may not follow the Gaussian distribution, for the commonly used SDEs such as the Variance Exploding SDE (VE) and Variance Preserving SDE (VP), the solution can be explicitly expressed as:

$$x_t = \alpha(t)x_0 + \beta(t)z, \tag{3}$$

where $z$ is sampled from $\mathcal{N}(0, I)$, and both $\alpha(t)$ and $\beta(t)$ are positive scalar functions. Using the forward SDE (2), we approach the task of data synthesis by gradually denoising the noisy observations $x_t$ to recover $x_0$, which is also accomplished through solving a reverse SDE. For the general formula and further details, readers can refer to [28, 32]. One fact we will use later is that the reverse SDE fundamentally relies on the score function $\nabla_x \log p_t(x)$, which serves as the parameterization objective in the score matching framework [28, 44].

**Example of discrete (cold) diffusion**    In contrast to the continuous heat diffusion framework, we now introduce a family of discrete Markov Chains as an instance of discrete (cold) diffusion. This example is particularly suitable for modeling categorical data. In the unconditional setting, the transition probability of the discrete diffusion at time step $t$ is defined as:

$$q(x_t|x_{t-1}) = \text{Multinomial}(x_t, p = x_{t-1}Q_t), \tag{4}$$

where $Q_t$ represents a Markov transition matrix. An illustrative example is the **absorbing diffusion**, as introduced by [30], where

$$Q_t = (1 - \beta_t)I + \beta_t e_m^T,$$

and $e_m$ is a one-hot vector with a value of 1 at the absorbing state $m$ and zeros elsewhere. Typically, the state $m$ is chosen as the 'masked' token. Hence, absorbing diffusion can be regarded as a masking process with a dynamical masking ratio $\beta_t$. Similarly to the heat diffusion, the corresponding reverse process can also be formulated as a discrete diffusion. However, the training objective shifts from fitting the score function (which only exists in the continuous case) to directly fitting the 'reconstruction probability' $p(x_0|x_t)$.

## 2.2   Graph-based molecular representation

In this paper, our primary focus is on the graph representation of molecules. Let $G = (V, E, P, H)$ denote the integrated molecular graph, consisting of $n := |V|$ atoms and $|E|$ bonds. The matrix $P \in \mathbf{R}^{n \times 3}$ is utilized to represent the 3D conformer of the molecule, containing the positions of each node. Moreover, within the graph $G$, the edge collection $E$ represents the graphical connections and chemical bond types between atoms. Additionally, we have the node-wise feature matrix $H \in \mathbf{R}^{n \times h}$, where, for the purposes of this article, we consider the formal charges and atom types as the components of $H$. In the method section, we will demonstrate how to build trajectories based on $G$.

## 3   Method

Now we illustrate the process of obtaining molecule augmented trajectories, based on which, MoleculeJAE is proposed to estimate the trajectory distribution. In Section 3.1, we will outline the construction of equivariant molecular trajectories. Subsequently, in Section 3.2, we will introduce our theoretical self-supervised learning framework for these trajectories. Our hypothesis is that a good representation should encode the information from the distribution of augmented trajectories. This hypothesis forms the practical basis of our core reconstructive and contrastive loss, which will be discussed in Section 3.3. Finally, in Section 3.4, we will present the key architectures. A comprehensive discussion of the related works is in Appendix B.

### 3.1   Equivariant Molecule Trajectory Construction

In the field of molecular representation learning, our objective is to jointly estimate the distribution of a molecule's 2D topology (including atom types and chemical bonds) and its 3D geometries (conformers). Building upon the notations in Section 2.2, our goal is to construct **augmented** trajectories $x_t := (H(t), E(t), P(t))$ from $G$. The challenging aspect lies in preserving the $SE(3)$ symmetry of the position matrix $P$, which we will describe in detail below.

Given two 3D point clouds (molecular conformers) $P_1$ and $P_2$, we say they are SE(3)-isometric if there exists an $R \in SE(3)$ such that $P_1 = RP_2$. In probabilistic terms, let $p_{3D}(x_{3D})$ be the probability density of all 3D conformers denoted by $\mathcal{C}$. Isometric conformers necessarily have the same density, i.e.,

$$p_{3D}(x_{3D}) = p_{3D}(\mathbf{R}x_{3D}), \quad \forall \mathbf{R} \in SE(3), x_{3D} \in \mathcal{C}. \tag{5}$$

Utilizing the symmetry inductive bias has been shown [24, 45] to greatly reduce the complexity of data augmentation. Additionally, in order to respect the $SE(3)$ symmetry, the augmented trajectory $P(0) \to P(t)$ should also be SE(3)-equivariant, satisfying

$$\mathbf{R}P(x_{3D}) = P(\mathbf{R}x_{3D}), \quad \forall \mathbf{R} \in SE(3).$$

This condition imposes a rigorous restriction on the form of the forward SDE (2) when it is applied to generate the augmented trajectory of conformers. However, if we constrain the form of $x_t$ to Eq. 3, the $SE(3)$ equivariance is automatically satisfied due to the $SE(3)$ equivariance of both the Gaussian random variable $z$ and the original data $x_0$. We leave the formal proof in Appendix A.

Regarding the 2D component, let $x_{2D}(t) := (H(t), E(t))$, where $x_{2D}$ consists of invariant scalars that remain unchanged under $SE(3)$ transformations. Although $x_{2D}$ is categorical in nature, we can treat them as continuous scalars that can be perturbed by Gaussian noise. During inference (generation), these continuous scalars are quantized back into categorical integers (see [22] for details). This allows both $x_{2D}(t)$ and $P(t)$ to undergo heat diffusion. By combining the 2D and 3D parts, we introduce the system of SDEs for $x_t$:

$$\begin{cases} dP(t) = -P(t)dt + \sigma_1(t)dw_t^1, \\ dH(t) = \mu_2(H(t), t)dt + \sigma_2(t)dw_t^2, \\ dE(t) = \mu_3(E(t), t)dt + \sigma_3(t)dw_t^3. \end{cases} \qquad (6)$$

It is worth mentioning that although the components of Eq. 6 are disentangled, the corresponding reverse diffusion process and its score function are entangled. A common choice of $\mu(x, t)$ is $\mu(x, t) := -x$, then will utilize the explicit solution of equations (6) to generate the molecular augmentation trajectory. Additionally, it is also possible to perform equivariant diffusion of the 2D and 3D joint representation through equivariant cold diffusion, following the approach described in (4). The detailed framework for this approach is provided in the Appendix A.

## 3.2 Equivariant Molecule Trajectory Learning with Density Auto-Encoding

To provide a formal foundation for self-supervised learning from augmented trajectories, we revisit the concept of denoising from a trajectory perspective. According to the Kolmogorov extension theorem [46], every stochastic process defines a probabilistic measure on the trajectory space, uniquely determined by a set of finite-dimensional joint distributions that satisfy consistency conditions. Therefore, estimating the probability of the infinite-dimensional trajectory space is equivalent to estimating the joint probabilities $p(x_{t_1}, \ldots, x_{t_k})$ for each finite time sequence $t_1, \ldots, t_k \in [0, T]$. From the standpoint of data augmentation, our specific focus is on learning the joint distribution of trajectory augmentation pairs: $p(x_0, x_t)$ (solution of Eq. 6), which determines how the noisy $x_t$ corresponds to the original (denoised) $x_0$.

To differentiate our notion of "Auto-Encoding" from the "denoising" method utilized in DDPM [32] (denoising diffusion probabilistic models), it is important to highlight that traditional diffusion-based generative models are based on marginal distribution modeling. In this framework, joint distributions induce marginal distributions, but not vice versa. This distinction becomes more apparent in the continuous SDE formalism of diffusion models [28], where multiple denoising processes from $x_t$ to $x_0$ can be derived while still sharing the same marginal distributions. However, the joint distributions of a probabilistic ODE flow significantly differ from those of the reverse SDE (as shown in Eq. 11) due to the deterministic nature of state transitions in an ODE between timesteps. In the following, we formally demonstrate this observation from the perspective of maximizing the trajectories' joint probabilistic log-likelihood.

For a given time $t \in [0, T]$, let us assume that the probability of the random pair $(x_0, x_t)$ defined in Eq. 1 is determined by a joint density function $p(x_0, x_t)$. Our objective is to approximate $p(x_0, x_t)$ within a variational class $p_\theta(x_0, x_t)$. Maximizing the likelihood of this joint distribution leads to the following optimization problem:

$$\text{argmax}_\theta \prod_{i=1}^n p_\theta(x_0^i, x_t^i), \qquad (7)$$

where $\{(x_0^i, x_t^i)\}_{i=1}^n$ represents the collection of $n$ augmented samples from the training dataset. Following the tradition of Bayesian inference [47], we parameterize $p_\theta$ by defining a joint energy function $\mathbf{E}_\theta(x_0, x_t)$ such that: $p_\theta(x_0, x_t) = \frac{1}{Z_\theta} e^{-\mathbf{E}_\theta(x_0, x_t)}$, where $Z_\theta(t)$ is the intractable normalization constant depending on $t$. Therefore, the maximal likelihood framework reduces to solving

$$\text{argmin}_\theta \mathbb{E}_{p(x_0, x_t)} \left[ \mathbf{E}_\theta(x_0, x_t) \right].$$

However, directly optimize $p_\theta(x_0, x_t)$ by taking the gradient with respect to $\theta$ is challenging because $\frac{\partial \log p_\theta(x_0, x_t)}{\partial \theta}$ contains an intractable term: $\frac{\partial Z_\theta(t)}{\partial \theta}$. To circumvent this issue, we borrow ideas from [48, 49] by treating the transformed $x_t$ as an infinitely dimensional "label" of $x_0$. More precisely, we consider the parameterized marginal density $q_\theta(x_0)$ as:

$$q_\theta(x_0) := \int p_\theta(x_0, x_t)dx_t, \qquad (8)$$

which involves integrating out $x_t$. We define the marginalized energy function with respect to $x_0$ as: $\bar{\mathbf{E}}_\theta(\cdot) := -\log \int \exp(-\mathbf{E}_\theta(\cdot, x_t))dx_t$. Now, let $f_\theta(x_0, x_t)$ denote the normalized conditional density function $p_\theta(x_t|x_0)$, we have $\frac{\partial \log f_\theta(x_t|x_0)}{\partial \theta} = -\frac{\partial \mathbf{E}_\theta(x_0, x_t)}{\partial \theta} + \frac{\partial \bar{\mathbf{E}}_\theta(x_0)}{\partial \theta}$. By taking the empirical expectation with respect to the finitely sampled pair $(x_0, x_t) \sim p(x_0, x_t)$, we can decompose the gradient of the maximum likelihood as follows (see Appendix A for the full derivation):

$$\tilde{\mathbb{E}}_{p(x_0,x_t)}\left[\frac{\partial \log p_\theta(x_0, x_t)}{\partial \theta}\right] = \tilde{\mathbb{E}}_{p(x_0)}\left[\frac{\partial \log q_\theta(x_0)}{\partial \theta}\right] + \tilde{\mathbb{E}}_{p(x_0,x_t)}\left[\frac{\partial \log f_\theta(x_0, x_t)}{\partial \theta}\right], \quad (9)$$

here we use $\tilde{\mathbb{E}}$ to denote the expectation with respect to the empirical expectation. Note that this decomposition holds for $(x_s, x_t)$ for any two different time steps ($0 \le s, t \le T$) of the trajectories. In the next section, we decompose Equation (16) into two sub-tasks, leading to our goal of optimizing these two parts simultaneously, as discussed below.

### 3.3 Reconstructive and Contrastive Tasks

In what follows, we denote the latent representation of data $x$ by $h_\theta(x)$ (the equivariant model for building $h_\theta$ will be discussed in the next section). Based on Eq. 16, we introduce two tasks for training $h_\theta(x)$ that stem from this decomposition.

**Reconstructive task.** The first term $\tilde{\mathbb{E}}_{p(x_0)}\left[\frac{\partial \log q_\theta(x_0)}{\partial \theta}\right]$ in Eq. 16 aims to reconstruct the distribution of data samples. Therefore, we refer this term as the **reconstruction** task, which involves modeling the marginal distribution $p_{data}(x_0)$ using $q_\theta(x_0)$.

Although it is possible to directly train the likelihood reconstruction term in the auto-regressive case, such as using the noise conditional maximum likelihood loss $\mathbb{E}_{t\sim[0,T]}\mathbb{E}_{x_t\sim p_t}\log q_\theta(x_t)$ proposed in [50], we instead adopt trajectory score matching [28], which is applicable for non-autoregressive graph data. The score matching loss is defined as follows:

$$\mathcal{L}_{sc} := \mathbb{E}_{t\sim[0,T]}\mathbb{E}_{p(x_0)p(x_t|x_0)}[\|\nabla \log p(x_t|x_0) - s_\theta(x_t, t)\|^2]. \quad (10)$$

We choose this approach for two reasons: First, training the score function enables the model to generate new samples by solving the reverse process, thus facilitating generative downstream tasks. Second, when $t = 0$, the score function for 3D structures can be interpreted as a "pseudo" force field for the molecular system [6], containing essential information about the molecular dynamics. Furthermore, [44] provided a rigorous proof demonstrating that score matching formula 10 serves as a variational lower bound for the marginal log-likelihood $\log q_\theta(x_0)$. This theoretical guarantee solidifies the effectiveness of score matching as a training objective.

For molecule trajectories, the score function encompasses both the 2D and 3D components. Moreover, the SE(3)-invariant density $p_{3D}$ defined by Eq. 5 implies that the corresponding 3D score function $\nabla_x p_{3D}(x)$ (represented as score($P_t$) in Fig. 2) is equivariant under $SE(3)$:

$$\nabla_x p_{3D}(\mathbf{R}x) = \mathbf{R}\nabla_x p_{3D}(x), \quad \forall \mathbf{R} \in SE(3), \ x \in \mathcal{C}.$$

In conclusion, the symmetry principle mandates that the score neural network takes an equivariant vector field (representing the positions of all atoms) and invariant atom features as input. It then produces two score functions that adhere to different transformation rules: 1. $\nabla_x p_{3D}(x)$: SE(3)-equivariant; 2. $\nabla_y p_{2D}(y)$: SE(3)-invariant.

**Contrastive task.** We have demonstrated that optimizing the score matching task allows us to capture information about the marginal distributions of the trajectories. However, the joint distribution contains additional valuable information. As an example, consider the following parameterized random processes, all of which share the same marginal distributions but exhibit varying joint distributions (as proven in [44]):

$$dy_t = [f(y_t, t) - \frac{1 + \lambda^2}{2}g^2(t)s_\theta(y_t, t)]dt + \lambda g(t)dB_t, \quad (11)$$

for $\lambda > 0$. Hence, it is theoretically necessary to optimize the second term $\tilde{\mathbb{E}}_{p(x_0,x_t)}\left[\frac{\partial \log f_\theta(x_0,x_t)}{\partial \theta}\right]$ of Eq. 16. By employing the conditional probability formula, we have:

$$f_\theta(x_0, x_t) = \frac{p_\theta(x_0, x_t)}{\int p_\theta(x_0, y)dy}. \quad (12)$$

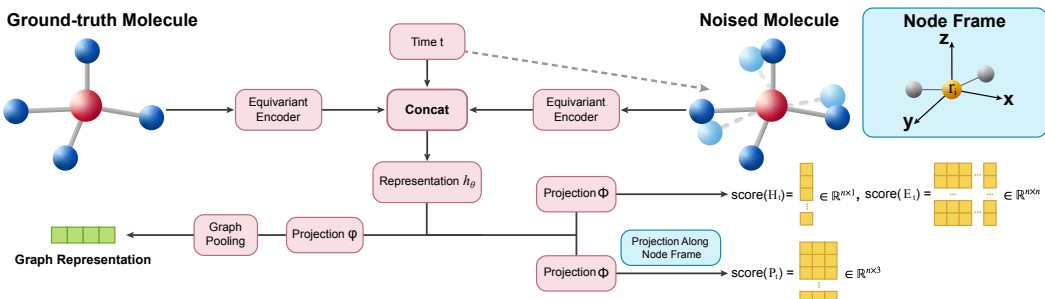

Figure 2: Architecture of MoleculeJAE. The inputs are ground-truth molecules with both 2D and 3D structures. MoleculeJAE also adopts the noised inputs denoising, so as to model the trajectory distribution. The outputs are three score functions for conformer and bond representations, which flow into the general pipeline in Figure 1.

It is important to note that the troublesome normalizing constant $Z_\theta(t)$ is cancelled out in Eq. 12. In practice, the integral $\int p_\theta(x_0, y) dy$ is empirically approximated using Monte Carlo sampling. To make a connection with contrastive learning (CL), recall that in CL, a common approach is to align the augmented views $((x, x^+))$ of the same data and simultaneously contrast the augmented views of different data $(x^-)$. By treating the joint distribution as a similarity measure, Eq. 12 can be seen as implicitly imposing two regularization conditions on $f_\theta(x_0, x_t)$: ensuring a high probability for $p(x_0, x_t)$ and a low probability for $p(x_0, y)$. This notion of similarity motivates us to refer to maximizing the second term of Eq. 16 as a **contrastive** task.

**Contrastive surrogate**   However, estimating $p_\theta(x_0, x_t)$ is challenging due to the intractability of closed-form joint distributions. To overcome this difficulty, we propose a black-box surrogate $\varphi$ that represents the mapping from the latent representation $h_\theta(x_t)$ to $p_\theta(x_0, x_t)$ (following the notation in figure 2). Specifically, let $(h_\theta(x_0), h_\theta(x_t))$ denote the representation pair obtained from the input molecule data $(x_0, x_t)$. Then, the surrogate of $p_\theta(x_0, x_t)$ is defined by $p_\theta(x_0, x_t) = \frac{1}{Z(t)} \exp\{-\frac{\|\varphi(h_\theta(x_0)) - \varphi(h_\theta(x_t))\|^2}{\tau^2(t)} + C(x_0)\}$. Here, $\tau(t)$ is a monotone annealing function with respect to $t$. By using Eq. 12, the unknown normalization constant $Z(t)$ and $C(x_0)$ cancel out, resulting in the following approximation:

$$f_\theta(x_0, x_t) \approx \frac{\exp\{-\|\varphi(h_\theta(x_0)) - \varphi(h_\theta(x_t))\|^2\}}{\int \exp\{-\|\varphi(h_\theta(x_0)) - \varphi(h_\theta(y))\|^2\} dy}. \tag{13}$$

Our surrogate is reasonable because our modeling target $p(x_0, x_t)$ is derived from the joint distributions of a (continuous) Markov process. When $x_t$ lies along the augmented trajectory of a specific data sample $x_0$ and approaches $x_0$ as $t \to 0$, the log-likelihood $\log p_\theta(x_0, x_t)$ is also achieves a local maximum. Therefore, based on the Taylor expansion, the leading non-constant term takes the form of a quadratic expression.

**Final Objective**   By combining the reconstruction and contrastive regularization tasks, we define the final **multi-task** training objective of MoleculeJAE as a weighted sum of the two tasks:

$$\lambda_1 \mathcal{L}_{sc} + \lambda_2 \mathcal{L}_{co}, \tag{14}$$

where $\lambda_1, \lambda_2$ are weighting coefficients, and $\mathcal{L}_{co} := \mathbb{E}_{t \sim [0,T]} \mathbb{E}_{(x_0, x_t) \sim p(x_0, x_t)} f_\theta(x_o, x_t)$. In Fig. 2, the noised input $x_t$ undergoes an encoding process to obtain the latent representation $h_\theta(x_t)$, which is then split into two branches:

1. The first branch passes through a score neural network $\phi$ for reconstruction: $s_\theta(x_t) = \phi(h_\theta(x_t))$;

2. The second branch incorporates the original data representation $h_\theta(x_0)$ and further projects the pair $(h_\theta(x_0), h_\theta(x_t))$ through a non-linear projection head $\varphi$ for contrastive learning.

Note that the black-box projection function $g$ (although not used in downstream tasks) also participates in the pretraining optimization, following the conventional contrastive learning frameworks [51, 52]. See Fig. 1 for a graphical illustration of MoleculeJAE's pipeline.

Table 1: Results on 12 quantum mechanics prediction tasks from QM9. We take 110K for training, 10K for validation, and 11K for testing. The evaluation is mean absolute error, and the best and the second best results are marked in **bold** and **bold**, respectively.

| Pretraining | $\alpha\downarrow$ | $\nabla\mathcal{E}\downarrow$ | $\mathcal{E}_{\text{HOMO}}\downarrow$ | $\mathcal{E}_{\text{LUMO}}\downarrow$ | $\mu\downarrow$ | $C_v\downarrow$ | $G\downarrow$ | $H\downarrow$ | $R^2\downarrow$ | $U\downarrow$ | $U_0\downarrow$ | ZPVE$\downarrow$ |
|---|---|---|---|---|---|---|---|---|---|---|---|---|
| – (random init) | 0.060 | 44.13 | 27.64 | 22.55 | 0.028 | 0.031 | 14.19 | 14.05 | 0.133 | 13.93 | 13.27 | 1.749 |
| Type Prediction | 0.073 | 45.38 | 28.76 | 24.83 | 0.036 | 0.032 | 16.66 | 16.28 | 0.275 | 15.56 | 14.66 | 2.094 |
| Distance Prediction | 0.065 | 45.87 | 27.61 | 23.34 | 0.031 | 0.033 | 14.83 | 15.81 | 0.248 | 15.07 | 15.01 | 1.837 |
| Angle Prediction | 0.066 | 48.45 | 29.02 | 24.40 | 0.034 | 0.031 | 14.13 | 13.77 | 0.214 | 13.50 | 13.47 | 1.861 |
| 3D InfoGraph | 0.062 | 45.96 | 29.29 | 24.60 | 0.028 | 0.030 | 13.93 | 13.97 | 0.133 | 13.55 | 13.47 | 1.644 |
| GeoSSL-RR | 0.060 | 43.71 | 27.71 | 22.84 | 0.028 | 0.031 | 14.54 | 13.70 | **0.122** | 13.81 | 13.75 | 1.694 |
| GeoSSL-InfoNCE | 0.061 | 44.38 | 27.67 | 22.85 | **0.027** | 0.030 | 13.38 | 13.36 | **0.116** | 13.05 | 13.00 | 1.643 |
| GeoSSL-EBM-NCE | 0.057 | 43.75 | 27.05 | 22.75 | 0.028 | 0.030 | 12.87 | 12.65 | 0.123 | 13.44 | 12.64 | 1.652 |
| 3D InfoMax | 0.057 | **42.09** | 25.90 | 21.60 | 0.028 | 0.030 | 13.73 | 13.62 | 0.141 | 13.81 | 13.30 | 1.670 |
| GraphMVP | 0.056 | **41.99** | 25.75 | **21.58** | **0.027** | 0.029 | 13.43 | 13.31 | 0.136 | 13.03 | 13.07 | 1.609 |
| GeoSSL-DDM-1L | 0.058 | 42.64 | 26.32 | 21.87 | 0.028 | 0.030 | 12.61 | 12.81 | 0.173 | 12.45 | 12.12 | 1.696 |
| GeoSSL-DDM | 0.056 | 42.29 | **25.61** | 21.88 | **0.027** | 0.029 | 11.54 | **11.14** | 0.168 | 11.06 | **10.96** | 1.660 |
| MoleculeJAE | **0.056** | 42.73 | 25.95 | **21.55** | **0.027** | **0.029** | **11.22** | **10.70** | 0.141 | **10.81** | **10.70** | **1.559** |

Table 2: Results on eight force prediction tasks from MD17. We take 1K for training, 1K for validation, and 48K to 991K molecules for the test concerning different tasks. The evaluation is mean absolute error, and the best results are marked in **bold** and **bold**, respectively.

| Pretraining | Aspirin $\downarrow$ | Benzene $\downarrow$ | Ethanol $\downarrow$ | Malonaldehyde $\downarrow$ | Naphthalene $\downarrow$ | Salicylic $\downarrow$ | Toluene $\downarrow$ | Uracil $\downarrow$ |
|---|---|---|---|---|---|---|---|---|
| – (random init) | 1.203 | 0.380 | 0.386 | 0.794 | 0.587 | 0.826 | 0.568 | 0.773 |
| Type Prediction | 1.383 | 0.402 | 0.450 | 0.879 | 0.622 | 1.028 | 0.662 | 0.840 |
| Distance Prediction | 1.427 | 0.396 | 0.434 | 0.818 | 0.793 | 0.952 | 0.509 | 1.567 |
| Angle Prediction | 1.542 | 0.447 | 0.669 | 1.022 | 0.680 | 1.032 | 0.623 | 0.768 |
| 3D InfoGraph | 1.610 | 0.415 | 0.560 | 0.900 | 0.788 | 1.278 | 0.768 | 1.110 |
| GeoSSL-RR | 1.215 | 0.393 | 0.514 | 1.092 | 0.596 | 0.847 | 0.570 | 0.711 |
| GeoSSL-InfoNCE | 1.132 | 0.395 | 0.466 | 0.888 | 0.542 | 0.831 | 0.554 | 0.664 |
| GeoSSL-EBM-NCE | 1.251 | 0.373 | 0.457 | 0.829 | 0.512 | 0.990 | 0.560 | 0.742 |
| 3D InfoMax | 1.142 | 0.388 | 0.469 | 0.731 | 0.785 | 0.798 | 0.516 | 0.640 |
| GraphMVP | **1.126** | 0.377 | 0.430 | 0.726 | **0.498** | 0.740 | 0.508 | 0.620 |
| GeoSSL-DDM-1L | 1.364 | 0.391 | 0.432 | 0.830 | 0.599 | 0.817 | 0.628 | 0.607 |
| GeoSSL-DDM | **1.107** | **0.360** | **0.357** | 0.737 | 0.568 | 0.902 | **0.484** | **0.502** |
| MoleculeJAE | 1.289 | **0.345** | **0.365** | **0.613** | **0.498** | **0.712** | **0.480** | **0.463** |

## 3.4 Model Architecture of MoleculeJAE

In pursuit of a meaningful latent molecular representation, we design our joint auto-encoder model as a conditional diffusion based model, inspired by [38, 39, 53]. In our model, the condition does not come from labels, but rather an encoding of the ground-truth molecule. Specifically, to obtain the latent representation $h_\theta$ shown in Fig. 2, we implement two **equivariant** encoders that satisfy the $SE(3)$ symmetry proposed in Section 3.1. One encoder takes the original molecule as input, and its output is used as a condition to help the other encoder that encodes the noisy molecule for reconstruction. The only requirement for the architecture of the two encoders is that the output should be invariant. Therefore, any $SE(3)$ equivariant GNN [54] that outputs invariant scalars will suffice.

**Equivariant decoder.** With our representation $h_\theta$ which depends on $(x_0, x_t, t)$, the decoder part of MoleculeJAE is divided into two heads. One head is paired with node-wise SE(3) frames to match the $SE(3)$ equivariant score function, while the other head generates an $SE(3)$ invariant representation that is used for contrastive learning (see Fig. 2 for a complete illustration). Further details on the model design are left in Appendix A.

## 4 Experiment

### 4.1 MoleculeJAE Pretraining

**Dataset.** For pretraining, we use PCQM4Mv2 [55]. It extracts 3.4 million molecules from Pub-ChemQC [56] with both 2D topology and 3D geometry.

**Backbone models.** We want to highlight that MoleculeJAE is agnostic to backbone geometric GNNs. In this work, we follow previous works in using SchNet model [57] for 3D conformation. For the 2D GNN representation, we take a simple version by mainly modeling the bond information (details in Appendix D). For the SDE models [28] for generating the joint trajectories, we consider both VE and VP, as in Eq. 6.

**Baselines for 3D conformation pretraining.** Recently, few works have started to explore 3D conformation pretraining. For instance, GeoSSL [5] provides comprehensive baselines. The initial

Table 3: Ablation studies of contrastive loss term in MoleculeJAE. The ablation results are on QM9.

| Pretraining | $\alpha\downarrow$ | $\nabla\mathcal{E}\downarrow$ | $\mathcal{E}_{\text{HOMO}}\downarrow$ | $\mathcal{E}_{\text{LUMO}}\downarrow$ | $\mu\downarrow$ | $C_v\downarrow$ | $G\downarrow$ | $H\downarrow$ | $R^2\downarrow$ | $U\downarrow$ | $U_0\downarrow$ | ZPVE$\downarrow$ |
|---|---|---|---|---|---|---|---|---|---|---|---|---|
| $\lambda_2=0$ | 0.057 | 43.15 | 26.05 | 21.42 | 0.027 | 0.030 | 12.23 | 11.95 | 0.162 | 12.20 | 11.42 | 1.594 |
| $\lambda_2=0.01$ | 0.056 | 42.73 | 25.95 | 21.55 | 0.027 | 0.029 | 11.22 | 10.70 | 0.141 | 10.81 | 10.70 | 1.559 |
| $\lambda_2=1$ | 0.066 | 45.45 | 28.23 | 23.67 | 0.028 | 0.030 | 14.67 | 14.42 | 0.204 | 13.30 | 13.25 | 1.797 |

Table 4: Ablation studies of contrastive loss term in MoleculeJAE. The ablation results are on MD17.

| Pretraining | Aspirin$\downarrow$ | Benzene$\downarrow$ | Ethanol$\downarrow$ | Malonaldehyde$\downarrow$ | Naphthalene$\downarrow$ | Salicylic$\downarrow$ | Toluene$\downarrow$ | Uracil$\downarrow$ |
|---|---|---|---|---|---|---|---|---|
| $\lambda_2=0$ | 1.380 | 0.359 | 0.363 | 0.744 | 0.482 | 0.902 | 0.548 | 0.590 |
| $\lambda_2=0.01$ | 1.289 | 0.345 | 0.365 | 0.613 | 0.498 | 0.712 | 0.480 | 0.463 |
| $\lambda_2=1$ | 1.561 | 0.547 | 0.781 | 0.735 | 0.918 | 1.160 | 1.052 | 0.809 |

three baselines involve type prediction, distance prediction, and angle prediction, respectively aiming to predict the masked atom type, pairwise distance, and triplet angle. The next baseline is 3D InfoGraph. It is a contrastive SSL method and predicts whether the node- and graph-level 3D representation are for the same molecule. Last but not least, GeoSSL proposes a new SSL family on geometry: GeoSSL-RR, GeoSSL-InfoNCE, and GeoSSL-EBM-NCE are to maximize the MI between the conformation and augmented conformation using different objective functions, respectively. GeoSSL-DDM optimizes the same objective function using denoising distance matching. GeoSSL-DDM-1L [6] is a special case of GeoSSL-DDM with one layer of denoising.

**Baselines for 2D and 3D multi-modal pretraining.** Additionally, several works have utilized both 2D topology and 3D geometry modalities for molecule pretraining. Vanilla GraphMVP [7] utilizes both the contrastive and generative SSL, and 3D InfoMax [58] only uses the contrastive learning part.

In the following, we provide the downstream tasks for applying our pre-trained MoleculeJAE. The experiment on joint generation of the 2D and 3D structures of molecules are provided in Appendix F.

### 4.2 Quantum Property Prediction

QM9 [59] is a dataset of 134K molecules, consisting of nine heavy atoms. It contains 12 tasks, which are related to the quantum properties. Among 12 tasks, the tasks related to the energies are very important, *e.g.*, $U$ and $U_0$ are the internal energies at 0K and 298.15K, respectively. The other two energies, $H$ and $G$ can be obtained from $U$ accordingly. The main results are in Table 1. We can observe that MoleculeJAE can outperform 12 baselines on 9 out of 12 tasks. We want to highlight that these baselines are very competitive, and certain works (*e.g.*, GeoSSL) also model the 3D trajectory. Noticeably, MoleculeJAE can reach the best performance on four energy-related tasks.

### 4.3 Molecular Dynamics Prediction

MD17 [46] is a dataset on molecular dynamics simulation. It contains eight tasks corresponding to eight organic molecules, and the goal is to predict the forces at different 3D positions. The size of each task ranges from 48K to 991K, and please check Appendix C for details. The main results are in Table 2, and MoleculeJAE can outperform 12 baselines on 6 out of 8 tasks and reach the second-best for one of the remaining tasks.

### 4.4 Ablation Study on the Effectiveness of Contrastive Loss

As discussed in Equation (14), there is one important hyperparameter $\lambda_2$ controlling the contrastive loss in MoleculeJAE. We want to conduct an ablation study on the effect of this contrastive term. As shown in Tables 3 and 4, we consider three value for $\lambda_2$: 0, 0.01, and 1. $\lambda_2 = 0$ simply means that we only consider the reconstructive task, and its performance is very close to $\lambda_2 = 0.01$, *i.e.*, the optimal results reported in Tables 1 and 2. However, as we increase the $\lambda_2 = 1$, the performance degrades by a large margin. Thus, we would like to claim that the contrastive term in MoleculeJAE is comparatively sensitive, and we need to tune this hyperparameter carefully to obtain optimal results.

## 5 Conclusion

In this work, we introduce a novel joint self-supervised learning framework called MoleculeJAE, which is based on augmented trajectory modeling. The term "joint" in our framework has two implications: Firstly, it signifies that our method is designed to model the joint distribution of trajectories rather than solely focusing on the marginal distribution. Secondly, it indicates that the

augmented molecule trajectory incorporates both 2D and 3D information, providing insights into different aspects of molecule representation. While our proposed method has demonstrated the best empirical results on 15 out of 20 geometry-related property prediction tasks, there are still areas left for improvement, such as architecture design. Please refer to Appendix E for an in-depth discussion.

## Acknowledgement

Jiujiu Chen would like to thank for her Associate Researcher Ruidong Chen, and Professor Xiaosong Zhang in UESTC for their special support.

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

# A  Methodology Detail

## A.1  Details for Section 3.3

Following Eq. 8, we consider the 'marginal' distribution

$$q_\theta(x_0) := \int p_\theta(x_0, x_t) dx_t$$

by marginalizing out $x_t$. We define the corresponding energy function with respect to $x_0$ as

$$\bar{\mathbf{E}}_\theta(x_0) = -\log \int \exp(-\mathbf{E}_\theta(x_0, x_t)) dx_t.$$

From the definition of $\bar{\mathbf{E}}_\theta(x_0)$, the distribution $q_\theta(x_0)$ shares the same normalization constant with $q_\theta(x_0, x_t)$. Therefore we have

$$\frac{\partial \log f_\theta(x_0, x_t)}{\partial \theta} = -\frac{\partial \bar{\mathbf{E}}_\theta(x_0, x_t)}{\partial \theta} + \frac{\partial \bar{\mathbf{E}}_\theta(x_0)}{\partial \theta},$$

where $f_\theta(x_0, x_t)$ denotes the normalized conditional probability $q_\theta(x_t|x_0)$. By the conditional probability formula, it is obvious that: it's obvious that

$$\frac{\partial \log q_\theta(x_0, x_t)}{\partial \theta} = \frac{\partial \log f_\theta(x_0, x_t)}{\partial \theta} + \frac{\partial \log q_\theta(x_0)}{\partial \theta}. \tag{15}$$

By taking the empirical expectation with respect to the sampled pair $(x_0, x_t) \sim p(x_0, x_t)$, we find that the gradient of the Maximum Likelihood allows the following decomposition:

$$\tilde{\mathbb{E}}_{p(x_0, x_t)} \left[ \frac{\partial \log p_\theta(x_0, x_t)}{\partial \theta} \right] = \underbrace{\tilde{\mathbb{E}}_{p(x_0)} \left[ \frac{\partial \log q_\theta(x_0)}{\partial \theta} \right]}_{\text{Gradient of reconstruction}} + \underbrace{\tilde{\mathbb{E}}_{p(x_0, x_t)} \left[ \frac{\partial \log f_\theta(x_0, x_t)}{\partial \theta} \right]}_{\text{Gradient of contrastive learning}}. \tag{16}$$

Here, we use $\tilde{\mathbb{E}}$ to denote the expectation with respect to the empirical expectation. In this way, we find that maximizing the likelihood $p_\theta(x_0, x_t)$ is equivalent to solving:

$$\text{argmax}_\theta \left\{ \tilde{\mathbb{E}}_{p(x_0)} \left[ \frac{\partial \log q_\theta(x_0)}{\partial \theta} \right] + \tilde{\mathbb{E}}_{p(x_0, x_t)} \left[ \frac{\partial \log f_\theta(x_0, x_t)}{\partial \theta} \right] \right\}.$$

## A.2  Details for Section 3.4

In this section, we provide the details for the model design described in Section 3.4. Specifically, we illustrate how to obtain the latent representation $h_\theta$ from $x_0 = (P_0, H_0, E_0)$ and $x_t = (P_t, H_t, E_t)$ (the solution of Eq. 6 at time $t$). It is important to note that in standard diffusion generative models [28, 32], only $x_t$ is fed into the score neural network.

The 3D part $(H_0, E_0)$ and $(H_t, E_t)$ are transformed by two equivariant 3D GNNs, and we denote the invariant node-wise output representation by $f_0$ and $f_t$. Additionally, we embed the time $t$ (which is essential for trajectory learning) using Fourier embedding, following [28]:

$$\text{Emd}(t) := \mathbf{Fourier}(t).$$

We obtain the 3D node-wise representation by concatenating the Fourier embedding, $f_0$, and $f_t$:

$$\mathbf{Node}_{3D} = \mathbf{MLP}\left[\text{Emd}(t) \parallel f_0 \parallel f_t\right].$$

Up to this point, we haven't utilized the 2D information $(E_0, E_t)$. Unlike the node-wise 3D representation, we encode $(E_0, E_t)$ as a weighted adjacency matrix $W$:

$$W = \mathbf{MLP}\left[\text{Emd}(t) \parallel E_0 \parallel E_t\right].$$

Combing $\mathbf{Node}_{3D}$ and the adjacency matrix $W$, we implement a dense graph convolution neural network (GCN) to obtain the final (node-wise) latent representation $h_\theta$, following [22]:

$$h_\theta = \mathbf{GCN}\left(\text{Node}_{3D}, W\right).$$

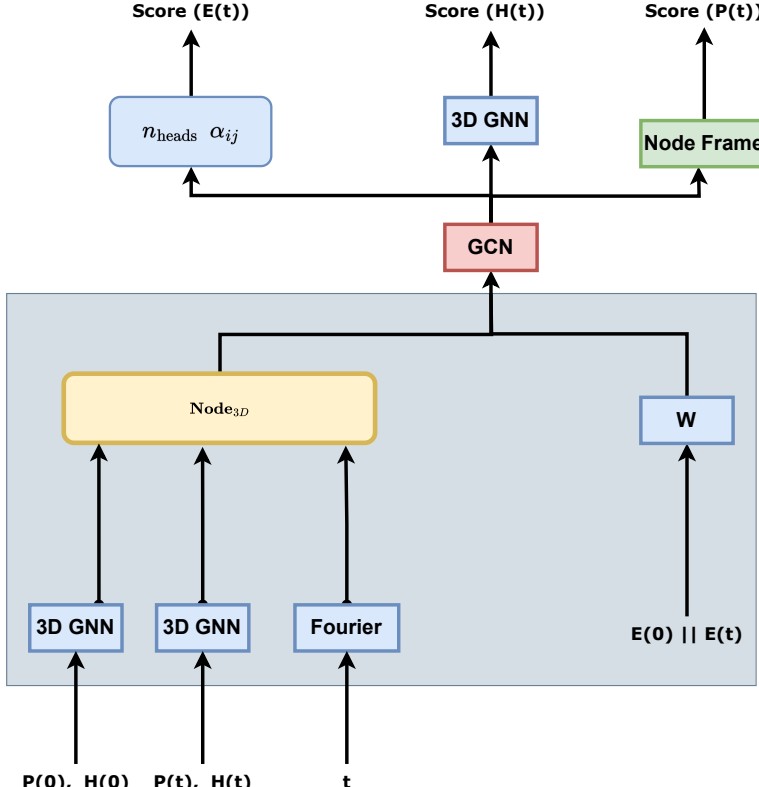

Figure 3: Modular Design of MoleculeJAE.

**3D output Score$(P_t)$.** To obtain the $SE(3)$ equivariant and reflection anti-equivariant vector field [60] Score$(P_t)$ from $h_\theta$, we implement the node-wise equivariant frame and perform tensorization, following [24, 54]. Consider node $i$ with 3D position $\mathbf{x}_i$, let $\bar{\mathbf{x}}_i := \frac{1}{N} \sum_{\mathbf{x}_j \in \mathcal{N}(\mathbf{x}_i)} \mathbf{x}_j$ be the center of mass around $\mathbf{x}_i$'s neighborhood. Then the orthonormal equivariant frame $\mathcal{F}_i := (\mathbf{e}_1^i, \mathbf{e}_2^i, \mathbf{e}_3^i)$ with respect to $\mathbf{x}_i$ is defined as:

$$\left( \frac{\mathbf{x}_i - \bar{\mathbf{x}}_i}{\|\mathbf{x}_i - \bar{\mathbf{x}}_i\|}, \frac{\bar{\mathbf{x}}_i \times \mathbf{x}_i}{\|\bar{\mathbf{x}}_i \times \mathbf{x}_i\|}, \frac{\mathbf{x}_i - \bar{\mathbf{x}}_i}{\|\mathbf{x}_i - \bar{\mathbf{x}}_i\|} \times \frac{\bar{\mathbf{x}}_i \times \mathbf{x}_i}{\|\bar{\mathbf{x}}_i \times \mathbf{x}_i\|} \right), \tag{17}$$

where $\times$ denotes the cross product, which is $SE(3)$ equivariant and reflection anti-equivariant. Then, we transform $h_\theta \in \mathbf{R}^{N \times L}$ to $h_{3D} \in \mathbf{R}^{N \times 3}$:

$$h_{3D} = (h_1, h_2, h_3) := \mathbf{MLP}(h_\theta).$$

Finally, for node $i$,

$$\text{Score}^i(P_t) = h_1 \cdot \mathbf{e}_1^i + h_2 \cdot \mathbf{e}_2^i + h_3 \cdot \mathbf{e}_3^i.$$

**2D output Score$(E_t)$.** Score$(E_t)$ represents the probability gradient with respect to the molecule bonds. Therefore, Score$(E_t)$ has the same shape as the dense adjacency matrix and is $SE(3)$ invariant. To obtain Score$(E_t)$, we leverage graph multi-head attention [61] to obtain a dense attention matrix $A_t$ (see [61] for the explicit formula):

$$A_t = \mathbf{Att}(h_\theta).$$

Then, we apply an **MLP** to the edge-wise $A_t$ to obtain the final 2D score function:

$$\text{Score}(E_t) = \mathbf{MLP}(A_t).$$

See Figure 3 for a graphical representation.

## A.3 A Cold Alternative for Section 3.1

Unlike continuous data that can take values ranging from $(-\infty, \infty)$, categorical data only takes a finite number of values. For example, in this paper, the atom type takes its value from the periodic table of elements, while the bond type takes its value from the set $\{0, 1, 2, 3, 4\}$. For these finite state spaces, we can replace our continuous diffusion framework with a discrete (cold) diffusion framework.

To define a discrete diffusion, we need to specify the transition matrix $Q_t$ for each time $t$, as stated in Eq. (4). Following [23, 29, 62], to effectively estimate the likelihood of the diffusion model, we further require that the noised state $q(z_t|x)$, defined as:

$$q(z_t|x) := xQ_1 \dots Q_t$$

to be equivariant under permutation. This requirement can be satisfied if we diffuse **separately on each node and edge feature**. Additionally, we require that the limiting distribution as $t \to \infty$ should not depend on the original data $x$, so that we can use it as a prior distribution for inference. A common choice is a uniform transition over the classes, where the corresponding $Q_t$ is defined as:

$$Q_t := \alpha_t \mathbf{I} + (1 - \alpha_t)\mathbf{1}_d\mathbf{1}_d^T/d$$

with $\alpha_t$ transitioning from 1 to 0. By letting $Q_t$ act on each node and each edge, we have defined the discrete diffusion for $H(t)$ and $E(t)$:

$$q(H_t|H_{t-1}) = H_{t-1} \cdot Q_t, \quad \text{and} \quad q(E_t|E_{t-1}) = E_{t-1} \cdot Q_t.$$

On the other hand, although the 3D structure is represented as a continuous point cloud, the molecule's conformer structures at the stationary states are discrete [5] and labeled by the energy levels of molecules. Through classical force field simulation, [63] provides a discrete set of rough 3D conformers (with their corresponding energies) for each molecule. Let $\{P_i\}_{i=1}^c$ be the collection of these rough 3D conformers, and denote uniform sampling on set $\{P_i\}_{i=1}^c$ as $\mathbf{Uniform}(\{P_i\}_{i=1}^c)$. Then, we can construct a discrete diffusion for the 3D structure as follows, similar to Eq. 3:

$$P(t) = \alpha(t)\mathcal{F}_0^{-1} \cdot P(0) + \beta(t)\mathbf{Uniform}(\{\mathcal{F}_i^{-1} \cdot P_i\}_{i=1}^c), \tag{18}$$

where $\mathcal{F}$ is a global equivariant frame obtained by averaging the node-wise equivariant frames $\mathcal{F}_i$ defined by Eq. 17. It can be verified that by projecting the original 3D structure with the inverse of its global frame $\mathcal{F}^{-1}$, the above equation is $SE(3)$ invariant.

Different from Eq. (3), where the randomness comes from Gaussian noise, the randomness in Eq. (18) arises from finite uniform sampling. In conclusion, we have constructed an $SE(3)$ invariant discrete 3D diffusion based on the ground truth and rough 3D conformers.

# B Related Works

The primary objective of self-supervised learning (SSL) is to acquire meaningful and compressed representations from unlabeled data, often measured by their ability to estimate a specific data distribution $p_{data}$. This leads to the hypothesis that a well-trained pretrained representation can enhance the robustness and generalization of downstream tasks. However, a challenging gap exists between the ground-truth distribution $p_{data}$ (if available) and our finite set of natural data samples. To address this gap, self-supervised learning techniques such as contrastive learning and manifold learning [64] often require additional data beyond the natural samples. Contrastive learning, for example, employs data augmentation to introduce positive and negative samples, while adversarial training [65] introduces adversarial samples through data perturbation for improving the robustness of downstream classification tasks.

**Adversarial and contrastive samples** In this work, we propose a diffusion framework for trajectory data augmentation based on the forward process of diffusion. A related concept that emphasizes the trajectory viewpoint is **consistency**, introduced in [66], which requires that the points on the backward path map to the same endpoint, corresponding to the condition of the trajectories' joint distribution. Additionally, the learned reverse (generative) process of a pretrained diffusion model has been employed to generate synchronized augmentation samples [42]. Adversarial samples [67], compared to data augmentation, are more subtle as they involve perturbations along specific directions of the

natural data that can be identified using the information provided by the score function $\nabla \log p_{data}(x)$ and $\nabla \log p_{data}(x|y)$ conditioned on labels $y$.

Once the augmented trajectories are established, our learning objective is to fit the joint distribution of the random trajectories. From a practical standpoint, our framework lies at the intersection of two unsupervised learning paradigms: contrastive learning and generative learning. In fact, [68] proposes an intriguing contrastive lower bound (positive sample part) for the reconstruction loss of a masked auto-encoder (AE). The authors further demonstrate that explicitly incorporating the negative sample part into the masked AE enhances the performance of the learned representation. Similarly, we observe a similar phenomenon within a more general unsupervised framework, as illustrated in Eq. 4, where the masking process is realized through cold diffusion.

**Molecular representation pretraining** The labels for molecules are scarce due to the laborious and expensive cost, and self-supervised pretraining has been widely adopted to alleviate this issue. For the **molecular topology pretraining**, existing unsupervised pretraining methods utilize either the reconstructing the masked subgraphs in molecules (AttrMask [1], GPT-GNN [3]) or detecting the positive and negative pairs in a contrastive manner (ContextPred [1], MolCLR [69]). Meanwhile, several works have studied the **molecular geometry pretraining**. GeoSSL [70] presents a comprehensive benchmark, including distance prediction, angle prediction, and an MI-based coordinate denoising framework specifically designed for conformations. Recent progress has started to combine these two research directions, *i.e.*, molecule **topology and geometry joint pretraining**. GraphMVP [7] first proposes to maximize the MI between these two modalities in a contrastive and generative manner, and 3D InfoMax [58] is a special case of GraphMVP by only considering the contrastive part. On the other hand, Unimol [71] introduced a novel 3D transformer for molecular pretraining, where their pretraining tasks involve reconstructing masked topological and geometric subgraphs in combination. The proposed MoleculeJAE in this work follows this research line.

**Alternative training objectives of diffusion models** In addition to the score matching training objective tailored for continuous probability distributions, an alternative training objective suitable for categorical probability distributions is the restoration loss proposed in [33]:[33]:

$$\min_{\theta} \mathbb{E}_t \mathbb{E}_{x_0 \sim p_{\text{data}}} \| R_\theta(D(x_0, t) - x_0 \|. \tag{19}$$

It is worth noting that compared to the score matching loss, Equation 19 is closer to the classical auto-decoder reconstruction loss if we disregard the time variable $t$. Furthermore, [31] has demonstrated that for discrete DDPM models, predicting $x_0$ and the score function are equivalent up to scaling constants, as the distribution of $x_{t-1}$ can be obtained in closed form through the forward process of DDPM using $x_0$ and the one-step noise.

Moreover, [72] proposed a reparameterization of the score matching loss, where the backbone neural network models the residual: $r_t = x_t - x_0$. Under the notation of Formula 3, the Soft Score Matching (SSM) in [72] is defined as:

$$\mathcal{L}_{ssm} := \mathbb{E}_{t \sim [0,T]} \mathbb{E}_{p(x_0)p(x_t|x_0)}[\|\alpha(t)(s_\theta(x_t, t) - r_t)\|^2].$$

Note that this reparameterization only works for the case when the closed form solution of the forward process has the form of Eq. 3.

## C Dataset Specification

Table 5: Some basic statistics on MD17.

| Pretraining | Aspirin ↓ | Benzene ↓ | Ethanol ↓ | Malonaldehyde ↓ | Naphthalene ↓ | Salicylic ↓ | Toluene ↓ | Uracil ↓ |
|---|---|---|---|---|---|---|---|---|
| Train | 1K | 1K | 1K | 1K | 1K | 1K | 1K | 1K |
| Validation | 1K | 1K | 1K | 1K | 1K | 1K | 1K | 1K |
| Test | 209,762 | 47,863 | 553,092 | 991,237 | 324,250 | 318,231 | 440,790 | 131,770 |

## D Implementation and Hyperparameter

In this section, we provide the detailed hyperparameters for implementing the experiments.

Table 6: Hyperparameter specifications for MoleculeJAE of property prediction.

| Hyperparameter | Value |
| --- | --- |
| epochs | {50, 100} |
| learning rate | {5e-4, 1e-4} |
| $\beta$ | {[0.1, 10]} |
| number of steps | {1000} |
| $\lambda_1$ | {1} |
| $\lambda_2$ | {0, 0.01, 1} |

Table 7: Hyperparameter specifications for MoleculeJAE of molecule generation.

| Hyperparameter | Value |
| --- | --- |
| epochs | {3000, 10000} |
| learning rate | {2e-4, 3e-4} |
| number of layers | 12 |
| number of diffusion steps | {500} |
| diffusion noise schedule | cosine |
| mlp hidden dimensions | {X: 256, E: 128, y: 128, pos: 64} |
| $\lambda_{train}$ | {X: 0.4, E: 2, y: 0, pos: 3, charges: 1} |

# E   Limitations

From an algorithmic standpoint, as briefly mentioned in Section 3.3, the parameterization of the score function for the reconstructive task can be obtained from the contrastive surrogate by numerically marginalizing the joint distribution $p_\theta(x_0, x_t)$ and subsequently applying automatic differentiation with respect to the data variable $x_0$. This methodology simplifies our two-head decoder framework described in Section 3.4 to a single head. Despite the additional computational cost associated with this approach, it holds value in terms of its theoretical significance and is therefore worth exploring in the future.

Another limitation is the treatment of the forward (noising) process of diffusion models as a form of trajectory augmentation in our current framework remains unparameterized. As the next step, it is important to parameterize the trajectory augmentation process similar to automatic data augmentation techniques [73, 74]. For instance, the flexible diffusion framework proposed in [29] could be adapted to the molecule graph setting, allowing for the joint optimization of molecule (forward + backward) trajectories. By incorporating such a parameterization, we can enhance the flexibility and effectiveness of our approach in modeling realistic molecule dynamics. This extension would allow for a more fine-grained control over the trajectory augmentation process and enable the optimization of molecule dynamics in a principled manner. Exploring these possibilities represents an exciting direction for future research.

# F   More Results

Table 8: Results on GEOM.

| Model | Mol stable ↑ | Atom stable ↑ | Validity ↑ | Unique ↑ | AtomTV ↓ | BondTV ↓ | ValW1 ↓ | Bond Lengths W1 ↓ | Bond Angles W1 ↓ |
| --- | --- | --- | --- | --- | --- | --- | --- | --- | --- |
| EDM (3000 epoch) | 5.5 | 92.9 | 34.8 | 100.0 | 0.212 | 0.049 | 0.112 | 0.002 | 6.23 |
| MiDi (2D+3D, 3000 epoch) | 69.2 | 99.0 | 67.4 | 100.0 | 0.059 | 0.024 | 0.036 | 0.012 | 5.47 |
| MoleculeJAE (Pretrained on PCQM4Mv2, 569 epoch) | **84.5** | **99.6** | **79.7** | **100.0** | **0.059** | **0.021** | **0.008** | **0.003** | **2.16** |

Table 9: Results on QM9.

| Model | Mol stable ↑ | Atom stable ↑ | Validity ↑ | Unique ↑ | AtomTV ↓ | BondTV ↓ | ValW1 ↓ | Bond Lengths W1 ↓ | Bond Angles W1 ↓ |
| --- | --- | --- | --- | --- | --- | --- | --- | --- | --- |
| EDM | 90.7 | 99.2 | 91.2 | 98.5 | 0.021 | 0.002 | 0.011 | 0.001 | 0.44 |
| MiDi (2D+3D, epoch 10000) | 94.83 | 99.66 | 95.38 | **97.4** | 0.023 | 0.006 | 0.008 | 0.007 | 1.407 |
| MoleculeJAE (2D+3D, Pretrained on PCQM4Mv2, epoch 3995) | **97.42** | **99.80** | **97.65** | 97.35 | **0.006** | **0.001** | **0.002** | **0.004** | **0.561** |

**2D + 3D molecule structure generation**   We evaluate MoleculeJAE's performance on unconditional molecule generation tasks. We generate both the graph structure and the conformer simultaneously, which is the joint distribution of the molecule's 2D and 3D structures. This task also gives us a chance to test if our MoleculeJAE pretraining framework is compatible with the (discrete) cold diffusion

framework3.1. The pretraining method still follows Figure 1 and 3, with the backbone 3D GNN of $(P(t), H(t))$ substituted by the MiDi transformer in [75]. The discrete diffusion formulas are the same as [75].

**Dataset**    We still apply the PCQM4Mv2 dataset as the pretraining dataset. For the unconditional molecule generation downstream dataset, we fine-tune our model on the QM9 dataset and the GEOM-DRUGS dataset [76]. GEOM comprises 430,000 drug-sized molecules with an average of 44 atoms 181 atoms. We follow [75] to split the dataset.

**Evaluation**    We test the generation performance by calculating the probability distance of 2D and 3D structures between our generated samples with the test set. The precise definition of these metrics are provided in section 5.1 of [75]. Besides the MiDi model in [75], we also use EDM [27] as our baseline. For our pretrained model, we finetune MoleculeJAE without the contrastive loss for 600 epoch in GEOM, and 4000 epoch in QM9.

**Results**    The experimental results are provided in Table 8 and 9, with our hyperparameters setting given in Table 7. We achieve state-of-art performance for both datasets.

