# OpenReview forum: "Molecule Joint Auto-Encoding: Trajectory Pretraining with 2D and 3D Diffusion"
_NeurIPS.cc/2023/Conference — NeurIPS 2023 poster_

### Official Review · Reviewer_oitq · 2023-06-24

**Soundness:** 3 good
**Presentation:** 3 good
**Contribution:** 3 good
**Rating:** 5
**Confidence:** 3

**Summary:**

The paper introduces a pretraining method for molecule joint auto-encoding (MoleculeJAE) for 2D molecular topology and 3D molecular geometry. Their approach adopts SE(3) symmetry and is trained by fitting the joint distribution of the trajectories from the forward process of the diffusion model.

The authors treat the 2D molecular structure and 3D geometry as continuous objects, which are perturbed by Gaussian noise in the diffusion process. Unlike traditional diffusion models that denoise data by marginal distribution modeling, their 'auto-encoding' approach focuses on learning the joint distribution of data pair $(x_0, x_t)$. The overall task is then optimized by two objective functions, including a reconstruction loss and a contrastive loss, as well as a surrogate model.

Experimentally, MoleculeJAE is pre-trained on the PCQM4Mv2 dataset and is used to perform downstream tasks on QM9 and MD17 datasets. The experimental results are competitive on property prediction and dynamics prediction tasks.

Overall, this work is well-motivated and novel, with a theoretical justification for their approach. This paper presents good quality work.

**Strengths:**

First of all, this work is well-motivated, as jointly learning 2D and 3D molecular representations by diffusion models is still at the stage of development. MoleculeJAE can learn both 2D bond topology and 3D conformation geometry information and is designed to respect the SE(3) symmetry of molecule data and is trained by fitting the joint distribution of the data’s augmented trajectories extracted from the forward process of the diffusion model.

It is also novel that the joint 2D and 3D learning of diffusion models is trained in a self-supervised learning manner and optimized by a contrastive learning objective function. MoleculeJAE unifies both contrastive and generative learning approaches from a trajectory perspective, providing a versatile and powerful molecular representation that can be applied to various downstream applications.

Unlike traditional diffusion modeling, MoleculeJAE uses a contrastive learning paradigm to fit the trajectory of pairwise molecules, while also utilizing the reconstructive task to perform denoising for each individual molecule. By using contrastive learning, it allows the model to align the augmented views of the same data and simultaneously contrast the augmented views of different data. Also, ablation studies show the importance to have a contrastive loss in the task.


**Weaknesses:**

For methodology, the authors treat the 2D graph topology as a continuous object that is perturbed by Gaussian noise. I understand it is easier and reasonable to put the three things, atom features, bonds, and coordinates, under a single SDE framework. However, following previous works [1][2], describing the 2D graph topology as a discrete object with a discrete diffusion process is more reasonable, and their models show superior results.
Experimentally, MoleculeJAE shows better results with only small margins. However, the authors do not compare the number of model parameters, the overall training time, and the error bars (standard deviation) with existing methods, so it is unclear to me how well MoleculeJAE surpasses those methods.

[1] Vignac, Clement, et al. "DiGress: Discrete Denoising diffusion for graph generation." arXiv preprint arXiv:2209.14734 (2022).

[2] Hua, Chenqing, et al. "MUDiff: Unified Diffusion for Complete Molecule Generation." arXiv preprint arXiv:2304.14621 (2023).

**Questions:**

1. Jointly learning 2D and 3D molecular data is not completely underdeveloped. Previous works on diffusion models, [1][2], have shown that diffusion models can be used to learn 2D and 3D latent molecular representations effectively jointly. They perform better on molecular generation tasks than single-modal diffusion models, and it is proper to cite these works and mention the difference.

2. I wonder if the authors consider extending this auto-encoding framework to discrete graph topology, as a discrete representation and diffusion process is more reasonable for graph-like objects. If not, I would like to hear the reason.

3. I would like to see the error bars (standard deviation), the number of model parameters, and the total training time compared with different methods.

[1] Hua, Chenqing, et al. "MUDiff: Unified Diffusion for Complete Molecule Generation." arXiv preprint arXiv:2304.14621 (2023).

[2] Vignac, Clement, et al. "Midi: Mixed graph and 3d denoising diffusion for molecule generation." arXiv preprint arXiv:2302.09048 (2023).


Typo:
1. Line 257, the log-likelihood logp is also achieves $\rightarrow$ the log-likelihood logp also achieves

---

> ### Author Rebuttal · Authors · 2023-08-09
>
> Thank you for your thoughtful review and valuable feedback. Certain aspects of your concerns related to motivation and experiments have also been covered in the general response. Please also review the comprehensive response provided there for further clarifications.
>
> **Weaknesses**
>
> A：Thank you for your insightful comments regarding our methodology and the referenced works. As we have outlined in section 2.1 and the appendix, our approach is indeed compatible with discrete diffusions [1] [2]. Moreover, we have extended our 3D structure modeling to incorporate the discrete diffusion framework. For detailed information, kindly refer to Appendix A.3. We also want to emphsis that our framework is mainly for unsupervised representation learning, as a result, certain related works focused exclusively on generation tasks have been inadvertently excluded. We will add the two references as you suggested.
>
> In response to your concerns about experiments, we have introduced a **new generation downstream task grounded in discrete diffusions**, which serves as a supplement to our property prediction downstream task. For further details, please consult the last part of the general response.
>
> **Question 1**
>
> A: We will discuss the relation between the mentioned papers and our method in the revised version. Notably, our approach introduces a novel unsupervised molecule representation learning framework centered around trajectory modeling. This contrasts with the focus of [1] and [2], which primarily concentrate on generation tasks.
>
> **Question 2**
>
> A: Your inquiry about extending our auto-encoding framework to discrete graph topology is astute. We have indeed considered this extension, and Appendix A.3 provides comprehensive details. From a methodology standpoint, our approach inherently encompasses discrete trajectories, akin to the formulation in [1] and [2].  Furthermore, an additional experiment addressing discrete generation has been performed in the **general response**.
>
> **Question 3**
>
> A: Thank you for raising these questions, and they are important details. We have illustrated the details below.
> 1. On the standard deviations on geometric property prediction tasks, we want to kindly point out that the computational resources for running such standard deviations are huge. Concretely, SchNet is by far the fastest geometric model (in comparison to PaiNN and Equiformer), and it may take around 12 hours for one task. We have 13 (pretraining algorithms) * 12 (tasks) * 12 (GPU hours) * 5 (seeds) = 9.36K GPU hours or 290 GPU days. Such computational cost is huge and impossible for us to add during the rebuttal period. This is also why the existing geometric pretraining methods (as the baselines in this work) do not include the standard deviations.
> 2. The model parameters depend on the backbone model.  Roughly speaking, the numer of parameters is the sum of backbone model, the projection head, and the MLP for the time-embedding. For the experiment in the paper,  the number of model parameters is 1.353M.
> 3. The training time for different pretraining algorithms are listed below:
> | Pretraining Algorithm | min / epoch |
> | -- | -- |
> | Type Prediction | 8 |
> | Distance Prediction | 7 |
> | Angle Prediction | 8 |
> | GeoSSL-RR | 10 |
> | GeoSSL-InfoNCE | 10 |
> | GeoSSL-EBM-NCE | 11 |
> | GeoSSL-DDM-1L | 12 |
> | GeoSSL-DDM | 19 |
> | 3D InfoMax | 9 |
> | GraphMVP | 11 |
> | Molecule JAE- light | 17 |
> | Molecule JAE | 20 |
> Here, we take all possible ingredients of Molecule JAE (see Fig. 2) into account, and the diffusion steps are also taken to be the maximal.  **Reducing the noise schedule of the diffusion model in Molecule JAE will further decrease the pretraining time** (Molecule JAE- light for non-generative downstream tasks).
>
> **Typos**: Thank you for highlighting the typos. We will rectify the mentioned error in the revised version.
>
> [1] Hua, Chenqing, et al. "MUDiff: Unified Diffusion for Complete Molecule Generation."
>
> [2] Vignac, Clement, et al. "Midi: Mixed graph and 3d denoising diffusion for molecule generation."

---

> > ### Comment · Reviewer_oitq · 2023-08-14
> > **Response by Reviewer**
> >
> > I thank the authors for their response. I will keep my score for a borderline accept. I hope the authors can properly cite and mention [1][2] in their revised version.
> >
> > [1] Hua, Chenqing, et al. "MUDiff: Unified Diffusion for Complete Molecule Generation."
> >
> > [2] Vignac, Clement, et al. "Midi: Mixed graph and 3d denoising diffusion for molecule generation."

---

> > > ### Author Response · Authors · 2023-08-14
> > > **Response from authors**
> > >
> > > Thank you for your feedback and for taking the time to consider our revisions. As we still have time within the discussion period, if there are particular aspects of our paper that you feel we should concentrate on or enhance further, we are willing to incorporate your further suggestions.

---

### Official Review · Reviewer_s6Nt · 2023-07-06

**Soundness:** 2 fair
**Presentation:** 3 good
**Contribution:** 2 fair
**Rating:** 6
**Confidence:** 4

**Summary:**

The paper proposes MoleculeJAE, an auto-encoder for both 2D and 3D molecule diffusion trajectories. The model learns the trajectories jointly in a self-supervised manner. Empirically, MoleculeJAE achieves competitive results on property and force prediction benchmarks.

**Strengths:**

The joint diffusion of 2D and 3D is novel. It is widely concerned that point cloud diffusion is less aware of 2D information. I admire the authors' contribution to this problem. I would also thank the authors for providing a detailed theoretical analysis.

**Weaknesses:**

1. I did not really get the motivation of modeling diffusion trajectories. Personally, I think the intermediate states are less informative than x_0. Could you please further explain your idea, and provide some additional ablation results if possible, for it seems that the current ablation study shows that the role of the contrastive loss is not very significant in many cases? Thanks.
2. I believe the baselines of QM9 experiments are not state-of-the-art. Please consider this work: https://openreview.net/forum?id=tYIMtogyee


**Questions:**

There is another work that introduced timestep embeddings in their diffusion process: https://ojs.aaai.org/index.php/AAAI/article/view/25639/25411 What are the advantages of contrastive learning?

---

> ### Author Rebuttal · Authors · 2023-08-09
>
> Thank you for your thoughtful review and valuable feedback. Certain aspects of your concerns related to motivation and experiments have also been covered in the general response. Please also review the comprehensive response provided there for further clarifications.
>
> **Weaknesses: Motivation of Modeling Diffusion Trajectories and contrastive learning**
>
> 1. Motivation:  In Sections 2.1, 3.2, and 3.3, we furnish an in-depth understanding of our motivation. To expound briefly, Section 2.1 introduces a novel trajectory-based approach, inspired by real molecule dynamics and diffusion generative models, to serve as continuous data augmentation. We subsequently adhere to the established protocol of unsupervised representation learning, aiming to deduce a learning objective from the augmented trajectory data distribution. Our focus centers on modeling the collective (joint) distribution of these equivariant trajectories, ultimately culminating in a fusion of a distinct contrastive element and generative learning objectives, as outlined in Section 3.3. The theoretical underpinning is furnished in Section 3.2.
>
> Regarding your concern about the **informativeness of intermediate states**, we have addressed this aspect in Lines 95-96, highlighting that for small t values, xt remains proximal to x_0. This phenomenon also informs our decision to abstain from inducing white noise typical in diffusion generative models (in experiments, we keep the noise schedule to be small). When the trajectory aligns closely with the original data, we consider it an augmentation of the initial dataset (just like Gaussian bluring and color distrotion are treated as augmentation of CV data).  On the other hand, scine the trajectory is continuous, each amount of transformation between two successive steps is small, and we care more on the correlation of successive steps.
>
> 2. We acknowledge your observation regarding the incremental significance of the **contrastive loss** in our ablation study.  We believe adding more optimization tricks customized for contrastive learning  (like adding the momentum update, EMA) may improve the abaltion results. The relevant experiments are currently in progress. However, we want to emphsis that we have theoretically demonstrated that the **joint distribution** of the trajectories contain more information than **marginal distributions**, and we utilized contrastive learning to extract the additional information for achieving better molecular representation.
>
> **Weaknesses: State-of-the-Art Baselines**
>
> Res: We appreciate your recommendation to include state-of-the-art baselines in our QM9 experiments. In Line 290, we emphasize the versatility of our pretraining framework with respect to geometric GNN backbones. Our approach **encompasses a fair testing protocol where the backbone remains constant (Schnet), while diverse pretraining tasks are explored**, as evident in our experimental section. In response to your suggestion, we have already integrated the pretraining framework in https://openreview.net/forum?id=tYIMtogyee into our experimentation (marked as **GeoSSL-DDM-1L**). Please see the general response (point 2 of the experiments part) for more details.
>
> **Question on a related work**
>
> Res: We will add the related work (https://ojs.aaai.org/index.php/AAAI/article/view/25639/25411) involving timestep embeddings in the diffusion process. However, we wish to highlight a fundamental distinction between our representation learning context and the generative model focus of the referenced work. Our trajectory distribution is harnessed as an augmented data distribution, resulting in the derivation of a **combined contrastive and generative approach** from the joint trajectory distribution, as expounded in our learning objectives. While contrasting with prior works, we also emphasize the intrinsic value of contrastive learning as a way of extracting the additional information from the joint ditribution rather than the marginal distributions (**which cannot been down by just introducing timestep embeddings**), substantiating its necessity theoretically and through ablation studies.
>
> Sidenote: please see the general response for an additional downstream task of jointly 2D and 3D generation.

---

> > ### Comment · Reviewer_s6Nt · 2023-08-11
> > **Thank the authors for the rebuttal**
> >
> > I think my concerns are well-clarified by the authors. I am updating my rating from 4 to 6.

---

> > > ### Author Response · Authors · 2023-08-11
> > > **Thanks**
> > >
> > > Thanks for your quick response. I sincerely appreciate your time and effort in re-evaluating our work.

---

### Official Review · Reviewer_fgxq · 2023-07-07

**Soundness:** 3 good
**Presentation:** 2 fair
**Contribution:** 3 good
**Rating:** 5
**Confidence:** 3

**Summary:**

This paper proposes a new representation learning method for molecules using 2D and 3D structures. The joint distribution between original molecules and augmented molecules is decomposed into reconstructive and contrastive tasks. The proposed model, MolecularJAE, simultaneously tackles both tasks with the help of SE(3) equivariant GNN models. MolecularJAE is evaluated on quantum property prediction and molecular dynamics prediction and achieves a competitive result against baseline models.


**Strengths:**

- Diffusion over two different modalities hasn't been explored as far as I know.
- The decomposition of joint distribution into reconstructive and contrastive tasks is new and interesting.


**Weaknesses:**

- It is known that the 3D structure of a molecule follows certain physical rules. The diffusion process used in this paper does not account for the prior knowledge from the domain.
- The performance of the proposed model is worse than some recently proposed models, which hasn't been included in the experiment section. For example, the performance of Equiformer[1] and PaiNN[2] on QM9 is better than the proposed model across almost all tasks.
- The presentation of this paper can be improved further. For example, Figure 1 does not provide any meaningful information to readers. Figure 2 is also difficult to digest. Polishing these figures with additional details and proper explanations would improve the accessibility of the manuscript.
- There are too many typos throughout the main text and appendix.

[1] Liao, Yi-Lun, and Tess Smidt. "Equiformer: Equivariant graph attention transformer for 3d atomistic graphs." ICLR 2023.

[2] Schütt, Kristof, Oliver Unke, and Michael Gastegger. "Equivariant message passing for the prediction of tensorial properties and molecular spectra." ICML 2021.


**Questions:**

- Why does the energy function formulation (Eq 7) reformulate the maximum likelihood estimator to the one above line 196, even though the normalizing constant depends on theta?
- The reconstructive task with score matching is advertised as a tool for generating new samples (line 222). Are there any experiments on the generative perspective? Given that the gain from the contrastive task is not significant in the experiments, the score-matching model seems well modeling the underlying distribution of graphs.

**Limitations:**

I couldn't find any negative societal impact.

---

> ### Author Rebuttal · Authors · 2023-08-09
>
> We express our sincere gratitude for your meticulous examination and insightful feedback. Now we want to address your concerns and clarify misunderstandings in detail.
>
> **Weaknesses**
>
> 1. **Incorporating Physical Rules**:  We appreciate your suggestion to integrate such knowledge into our diffusion process. However, we must acknowledge the practical challenge in precisely determining the drift and diffusion coefficients for Eq.6. We've explored the avenue of parameterizing these coefficients, as per a flexible framework that paramterizing general physical diffusions [1], yet found the resultant improvement to be marginal. In response, we intend to incorporate the pertinent experimental outcomes into the appendix as a releted emperical study.
>
> 2. **Performance Analysis**:  In light of your observation, we emphasize our flexibility in employing various geometric GNN backbone models, as articulated in Line 290. Our pretraining framework operates agnostically to the choice of backbone models, including PaiNN and Equiformer. We contend that **evaluating a pretraining method fairly necessitates maintaining a fixed backbone model while testing diverse pretraining frameworks**, as demonstrated in our experimental section. While we are indeed conducting experiments with a PAINN backbone model [2] as per your suggestion (see the general response), we are confident that our current experiments substantiate the efficacy of our approach.
>
> 3. **Enhancing Clarity of figures and Correcting Typos**:  We recognize your concerns regarding Figure 1 and Figure 2. We perceive Figure 1 as illustrative of our novel trajectory modeling pipeline, encompassing both contrastive and denoising elements. Similarly, Figure 2 elucidates our novel joint 2D and 3D information encoding while preserving symmetry.
>
> We appreciate your feedback on typographical errors and will address them in line with other reviewers' suggestions. To expedite this process, we kindly request the specific locations of these errors.
>
> **Questions**
>
> 1. We appreciate your astute observation. To address this, we will incorporate the expectation of the normalization constant $E_x(Z_{\theta}(x))$ into Eq. 7 to accurately represent the energy function formulation. Note that this factor will disappear in Eq. 8, therefore doesn't affect the the correctness of the rest of the argument.
>
> 2. We've indeed explored the generative perspective through experiments in our ablation study. Moreover, we perform an additional generative task to our pretrained representation. As can be seen in Fig.2, the right-hand side of the model's output is ready for downstream generation tasks. Please consult the experiment section of the general response for details.
>
> [1] "A flexible diffusion model"  ICML 2023
>
> [2] "Equivariant message passing for the prediction of tensorial properties and molecular spectra"  ICML 2021.

---

> ### Author Response · Authors · 2023-08-16
> **Gentle Reminder**
>
> Hope this message finds you well.  We have carefully addressed your questions and have incorporated additional generative experiments (in the general response) based on your valuable advice. We are eager to know if our response has effectively addressed the concerns you raised in your initial review. Should you need further clarification or have any additional points you would like us to consider, please do not hesitate to share your thoughts. We are committed to ensuring that all your concerns are fully addressed.
>
> Thanks!

---

> > ### Comment · Reviewer_fgxq · 2023-08-17
> >
> > I appreciate your comprehensive response regarding my questions and concerns. Based on the rebuttal, I think my biggest concern about the paper on the experimental performance has been properly addressed. I hope the authors add the new results with missing details to the revised manuscript to make it more concrete. In this regard, I am happy to raise my original score.

---

> > > ### Author Response · Authors · 2023-08-17
> > > **Thanks**
> > >
> > > We sincerely appreciate your time and effort in re-evaluating our work. We will make the suggested revisions as promised in our updated manuscript.

---

### Official Review · Reviewer_LUAH · 2023-07-07

**Soundness:** 3 good
**Presentation:** 2 fair
**Contribution:** 3 good
**Rating:** 6
**Confidence:** 3

**Summary:**

The authors propose an auto-encoding method for learning molecular embeddings from both 3D and 2D information jointly. The method is loosely related to diffusion methods in that embeddings of data augmentation trajectories are learned via a score-based reconstruction loss and contrastive loss. The embeddings give SOTA performance on a number of QM and activity prediction tasks.

**Strengths:**

### Originality

The method is original in that it encodes both 3D and 2D molecular information via learning to embed data augmentation trajectories.

### Quality

The results indicate that the resulting embeddings are SOTA.

### Clarity

The paper could be made much clearer (see Weaknesses).

### Significance

Optimal molecular embeddings are a perennial desire for all kinds of downstream molecular tasks. The SOTA performance of these embeddings suggests that this method represents a significant contribution.

**Weaknesses:**

- Section 2 Background is a meandering presentation of a number of related ideas, some of which are not immediately relevant for the main paper. Those should be moved to the appendix, and the remainder should be made much crisper so that it is clear to the reader how the remaining sections will proceed.
  - the unnumbered equation after Eq. 4 is missing an $e_m$.
- It is not at all clear what data augmentations are used in this paper. E.g. what are $\mu_1$ and $\mu_2$ in Eq. 6?
- The presentation of the MLE in Section 3.2 is unnecessarily confusing. There is no need to introduce energy-based models, Eq. 9 can be derived without the energy functions. Note that the energy functions are never referred to again. The only thing is the surrogate gaussian KDE, but again that does not need to be motivated by energy functions.

Typos:
- The statement $E_θ(x0) = \log q_θ(x_0)$ is incorrect since it is missing a $\log Z_\theta$.
- Most of the score expressions in section 3.2 are missing a $\log$.

**Questions:**

- The unnumbered equation after Eq. 5 does not make sense - how does one apply a rigid transformation to time t?
- How are the results in Tables 1 and 2 obtained? What model is trained on the learned embeddings?
- What is the intuition behind this entire paper? Why does embedding a trajectory give better performance than embedding just the stationary data? How do these results depend on the choice of trajectories?

**Limitations:**

Some limitations are addressed in the appendix.

---

> ### Author Rebuttal · Authors · 2023-08-09
>
> We express our sincere gratitude for your meticulous examination and insightful feedback.
>
> **Weaknesses: Organization of Section 2 and Section 3.2 and Equation Missing $e_m$** : To rectify this, we will enhance the clarity of Section 2 by trimming the general introduction of the diffusion mechanism . Our intention is to directly establish a linkage between our equivariant trajectory approach and the classical data augmentation technique, employing the heat and discrete (cold) diffusion equations (3) and (4). The terminology "energy" at line 195-196 is employed solely to represent the logarithm of the probability density function, a convention consistent with Bayesian optimization literature.
>
> To response your concern on introducing the general energy based models,  We will streamline the MLE exposition in Section 3.2, omitting the energy-based models' introduction for Equation 9 derivation.
>
> Furthermore, to optimize space usage, we will introduce a comprehensive roadmap at the conclusion of the introduction. This addition will illuminate the forthcoming sections' structure, easing the reader's navigation through our paper.
>
> **Clarification on Data Augmentations in Eq. 6**: We deeply appreciate your scrutiny. We commit to providing unequivocal clarity by explicitly stating that $\mu(x,t) = -x$ for both $x=E(t)$ and $H(t)$ in Eq. 6. We also use a discrete equation of $E(t)$ and $H(t)$ (formula is given in appendix) for an additional gneration task suggested by other reviewers. Please concult the general response.
>
> **Addressing Typos on missing $\log$**:  We will delete the statement on $E_{\theta}(x_0)$. We want to emphasis that the rest of the  argument remains valid due to the cancellation of $Z_{\theta}$ by conditional probability definition. The assertion regarding missing "score" expressions in Section 3.2 seems a misunderstanding,  since no "score' appeared in section 3.2.
>
> **Questions**:
>
> Q1: The unnumbered equation following Eq. 5 will be rectified: t will be replaced with $x_{\text{3D}}(t)$.
>
> Q2: For results in Tables 1 & 2, they are obtained by re-running standard pretraining-and-finetuning results (the references are given in Line 295-304) in one V-100 GPU. Specifically:
> - In the pretraining stage, we pretrain an encoder (e.g., the equivariant encoder in Figure 1) using various pretraining algorithms.
> - In the finetuning stage, we add a linear prediction head on the pretrained encoder and then take an end-to-end finetuning.
>
> Q3: The paper's intuition is outlined in sections 2.1, 3.2, and 3.3. Essentially, we propose trajectory utilization (inspired by real molecule dynamics and diffusion generative models) for continuous data augmentation. Via standard unsupervised representation learning, we extract learning objectives from augmented data distributions. Our model captures joint trajectory distributions, culminating in a combination of contrastive and generative learning objectives in section 3.3, supported theoretically in section 3.2.  As to the question of  "How do these results depend on the choice of trajectories",  we recognize the merit in investigating this avenue further. It aligns intriguingly with the exploration of whether a **composition of diverse data augmentations (trajectory-based or not)** can enhance representation learning. Consequently, we intend to address this pertinent question in our upcoming "Future Outlook" section.

---

> ### Author Response · Authors · 2023-08-18
> **A Kind Reminder**
>
> I hope this message finds you in good health. I wanted to take a moment to express our sincere gratitude for your positive evaluation of the novelty and methodology of our paper, as well as your insightful suggestions for improving our presentation. Your feedback has been invaluable, not only in addressing critical typos that play a crucial role in ensuring readers' understanding of our method, but also in shaping the overall organization of our revisions.
>
> We have carefully considered your feedback and have made efforts to address the concerns you raised.  As we approach the conclusion of the discussion period, we are eager to ensure that our response effectively addresses your concerns and aligns with your expectations.   If there are any lingering questions or if you require further clarification on any aspect of our work, please don't hesitate to reach out to us. We are committed to ensuring that your concerns are fully addressed before the deadline.

---

### Author Rebuttal · Authors · 2023-08-08

## General Response
We thank all the reviewers for their time, and valuable feedback for improvements.  All relevant works and typos mentioned by reviewers will be discussed in the revised version. Now, we clarify and address some common issues that have been raised by the reviewers.

**Motivation and intuition** (Reviewer LUAH, s6nt, oitq): We appreciate the opportunity to clarify the motivation and intuition behind our approach. In **Section 2.1**, we introduced the concept of utilizing trajectories as continuous data augmentation, drawing inspiration from real molecule dynamics and diffusion generative models. Following the standard procedure of unsupervised representation learning, we derived a learning objective from the augmented data distribution. Our methodology involves modeling the joint distribution of trajectories (as elaborated in **Section 3.2**), which culminates in a novel combination of contrastive and generative learning objectives, outlined in **Section 3.3**.  In conclusion, **we learn a powerful representation by fitting the joint distribution of augmented equivariant trajectories which contain both 2D and 3D molecular information**. We hope this clarification better emphasizes the foundation and significance of our proposed approach.

**Paper organization** (LUAH, fgxq):  To enhance clarity, we will relocate certain content from the **background section** to the appendix, ensuring a smoother flow for readers. Additionally, we acknowledge the importance of providing a clear **roadmap** for our work. In response, we will incorporate a comprehensive roadmap in the final paragraph of our introduction. Furthermore, more detailed illustrations of the **figures** will be given within the main text.

**Experiments**  (fgxq, s6Nt, oitq): A fundamental misunderstanding is that our primary aim is to present a comprehensive unsupervised molecule representation learning framework rooted in equivariant symmetric trajectories. Therefore, the most straightforward way to demonstrate our method's effectiveness is to **fix an equivariant backbone neural network** (we choose Schnet), and test if the representation trained by our method can beat other representation learning methods.  In contrast, some **experiments suggested by reviewers (e.g., comparing our QM9 experimental results to PaiNN[1], GNS-TAT[2]) involve different research settings**, and we believe our core experiments have effectively showcased the efficacy of our approach.  To further address your suggestions, we have conducted three additional experiments (** only partial results have been shown due to severe GPU shortage in the rebuttal period**).

1. We performed experiments **replacing the Schnet backbone with PaiNN** on certain QM9 tasks, and our MoleculeJAE achieved state-of-the-art performance. **The table is given in the attached PDF**.

2. We want to highlight that this work [2] comprises a backbone model (GNS-TAT) and a pretraining algorithm (GeoSSL-DDM-1L). (1) The default GNS-TAT in [2] is not an equivariant neural network. (2) **We have already experimented GeoSSL-DDM-1L**. As shown in Tables 1 & 2 on page 8, we can observe that MoleculeJAE is better than GeoSSL-DDM-1L.

3. Reviewer oitq suggests us to test if our **pretrained representation performs well for generation task**. We apply our generalized (discrete) framework in Appendix A.3 for a joint 2D and 3D structure generation (the detailed setting is given in [3]), revealing significant enhancements over non-pretrained models:

Table 1: Results on GEOM-Drugs with explicit hydrogens. We pretrained our model MoleculeJAE on PCQM4Mv2 for 24 epochs and then finetuned the generative head (see Figure 2) of MoleculeJAE on GEOM-Drugs for 211 epochs. As the table shows, MoleculeJAE achieves SOTA results on 9 out of 10 metrics.

|Model|Mol stable↑|Atom stable↑|Validity↑|Unique↑|AtomTV↓|BondTV↓|ValW1↓|Bond Lengths W1↓|Bond Angles W1↓|
| ------------ | ------------ | ------------ | ------------ | ------------ | ------------ | ------------ | ------------ | ------------ | ------------ |
|EDM (3000 epoch)|5.5|92.9|34.8|100.0|0.212|0.049|0.112|0.002|6.23|
|MiDi (2D+3D, 3000 epoch)|69.2|99.0|67.4|100.0|0.059|0.024|0.036|0.012|5.47|
|MoleculeJAE (Pretrained on PCQM4Mv2)|84.5|99.6|79.7|100.0|0.059|0.021|0.008|0.003|2.16|

We believe these supplementary experiments underscore the robustness and versatility of our approach.

[1] Equivariant message passing for the prediction of tensorial properties and molecular spectra

[2] Pre-training via Denoising for Molecular Property Prediction

[3] Midi: Mixed graph and 3d denoising diffusion for molecule generation

---

> ### Author Response · Authors · 2023-08-21
> **A simple update**
>
> Following the reviewer's suggestion, we update the **Lumo** downstream task for our supplementary Painn pretraining model (missed in the last period of experiment due to  GPU shortage):
>
> | Model |  Lumo  |
> |  ----  | ----  |
> | no-pretraining |  20.16  |
> | Distance Prediction | 18.26 |
> | 3D InfoGraph | 20.16 |
> |GeoSSL-EBM-NCE | 17.89 |
> |3D InfoMax | 17.69 |
> |GraphMVP | 17.02 |
> | GeoSSL-DDM | 16.95 |
> | 3D-EMGP | 23.99 |
> |**Molecule-JAE** | **16.89** |

---

### Decision · Program_Chairs · 2023-09-21

**Decision:**

Accept (poster)

**Comment:**

The authors propose an auto-encoding method for learning molecular embeddings from both 3D and 2D information jointly. It achieves strong performance on a number of QM and activity prediction tasks. The presentation of the paper can be improved (make the related work more concise, removing MLE and energy-based model). Additional experiment comparing with Liao et al 2023, Shutt et al 2021, can be added.

Based on the reviews, I full support inclusion of this paper in the conference.